# Design of Ultra-High Extinction Ratio TM- and TE-Pass Polarizers Based on Si-Sc_0.2_Sb_2_Te_3_ Hybrid Waveguide

**DOI:** 10.3390/mi13040495

**Published:** 2022-03-23

**Authors:** Xuanxuan Xie, Furong Liu, Qingyuan Chen, Yongzhi Zhang

**Affiliations:** 1MOE Key Laboratory of Trans-Scale Laser Manufacturing Technology, Beijing University of Technology, Beijing 100124, China; xiexuanxuan@emails.bjut.edu.cn (X.X.); qychen@emails.bjut.edu.cn (Q.C.); zhangyz@bjut.edu.cn (Y.Z.); 2Beijing Engineering Research Center of Laser Technology, Beijing University of Technology, Beijing 100124, China; 3Institute of Laser Engineering, Faculty of Materials and Manufacturing, Beijing University of Technology, Beijing 100124, China

**Keywords:** TM- and TE-pass polarizers, Sc_0.2_Sb_2_Te_3_, high extinction ratio, phase-change material, integrated silicon photonics device, nanophononics

## Abstract

The selective polarizers play an important role in silicon-based integrated circuits. The previous polarizers based on silicon waveguides have the defects of large scale and low extinction ratio. In this work, TM- and TE-pass polarizers only 10 μm long were developed based on phase-change material of Sc_0.2_Sb_2_Te_3_ (SST) hybrid silicon waveguide, where several SST bars with a varied distance was designed. Because of the excellent characteristics of the refractive index of the material, ultra-high extinction ratios (ERs) were achieved. A 3-D finite element simulation was carried out to optimize the structure of the polarizers, and the distribution of light field, as well as the transmission behavior of TE and TM modes in the two polarizers, was further demonstrated in detail. When the SST is crystalline, the unwanted mode can be attenuated, while the wanted mode can pass through with low loss. Compared with the GST-based polarizers, the proposed ones achieved high extinction ratios of ~43.12 dB (TM-pass one) and ~44.21 dB (TE-pass one), respectively; at the same time, ILs for the wanted modes could be negligible. The design of high-performance polarizers paves a new way for applications of all-optical integrated circuits.

## 1. Introduction

Currently, silicon on insulator (SOI) platforms have been widely used in photonic devices due to the high refractive index difference between the cladding and the core. The light is bound in the core strongly to realize low loss transmission [1]. However, the birefringence caused by the high refractive index difference makes the device polarization-dependent, so, the polarization handling device is needed for the development of silicon (Si) photonics technology. Traditional polarizers based on silicon waveguides are mainly divided into two categories: (1) one is to separate or convert one of the modes into the required modes, such as polarizer splitter [2,3,4,5] and polarization rotator [6,7,8], but such devices have the disadvantages of large size and complex structure. (2) The other is to eliminate the unnecessary modes and retain the desired mode with a low loss output [9,10], but they are usually with small extinction ratios, which are not suitable for the application. Recently, some hybrid waveguides, consisting of phase-change materials (PCMs) integrated with SOI, were designed to achieve all-optical non-volatile selectivity of the modes with an ultra-small footprint, conducive to integration [11,12,13,14]. That is because PCMs, such as vanadium dioxide (VO_2_) [15,16,17] or germanium–antimony–tellurium alloys (Ge_2_Sb_2_Te_5_, Ge_1_Sb_2_Te_4_, etc.) [18,19,20], have a large contrast between two states of amorphous and crystalline [21,22,23], which can be applied to be a polarizer to meet the requirements of high extinction ratio and low loss. L.D. Sanchez et al. [14] reported a tunable 20 µm-long TE-pass polarizer based on hybrid Si-VO_2_ technology with an IL ~5.5 dB. However, compared with germanium–antimony–tellurium alloys, the phase transition in VO_2_ is volatile, namely requiring extra energy to maintain the states, leading to a higher energy loss. Therefore, non-volatile PCMs [24,25,26,27,28] are more promising. More recently, an active TE- and TM-pass polarizer based on Ge_2_Sb_2_Te_5_ (GST) in Si waveguide was demonstrated with a relatively high IL (~2.6 dB) [27]. Y.P. Song et al. [13] also proposed a low IL (<0.33 dB) active TE-pass polarizer on the SOI platform utilizing GST-assisted hybrid waveguide to attenuate the TM mode, however, the associated scale of the active region is relatively large (5.5 µm long). Moreover, the phase-change time (namely crystallization time) of germanium–antimony–tellurium alloys is commonly in the scale of nanoseconds, restricting some applications required in the scenarios of sub-nanosecond operations, such as ultrafast computing, storage, and switch, etc. Recently, a cache-type PCM of Sc_0.2_Sb_2_Te_3_ (SST) has been proposed [29], with the crystallization time one order of magnitude shorter than that of GST, more suitable for ultrafast photonics devices. However, it has only been reported in the application of electronic storage [30,31], and more applications are required to be explored, especially for some ultrafast photonic devices.

In this paper, we designed two novel polarizers of TM- and TE-passes based on SST array hybrid Si waveguides for integrated circuits, respectively. Additionally, the distribution of the modal, as well as the transmittance behavior in the two polarizers was further demonstrated in detail. Finally, the properties of the polarizers proposed were also compared with the GST-based ones.

## 2. Simulation Methods

### 2.1. Device Structure and Design

The schematic diagrams of the TM-pass polarizer based on SST and Si waveguide are presented in Figure 1a. The five bars of SST thin films are placed on the top of the waveguide. The Si waveguide with a fixed size of 10 µm long, 500 nm wide, and 340 nm high was designed to locate on the top of the SOI matrix. The width of the input silicon waveguide was chosen as W_si_ = 500 nm to ensure a single mode. Figure 1b shows the cross-section of the hybrid waveguide. The width of SST thin film is 500 nm. The length, thickness, and the number of the SST film are defined as L_sst_, H_sst_, and N. A factor of g_n_, defined as the gap between two adjacent SST films, is written as g_n_ = g_n−1_ + d, where, g_n_ and g_n−1_ are two adjacent gaps, and d is the increment of the gap. Considering the actual fabrication flexibility, we choose the first gap (g_1_) as g_1_ = 100 nm [32,33].

The TE-pass polarizer consists of a Si waveguide and SST films, the five bars of SST thin films are attached to both sides of the waveguide, as shown in Figure 1c,d. The width, length, and height of SST are W′_sst_, L′_sst_, and H′_sst_. The difference is two equal-sized SST arrays are clung to the left and right sides of the Si waveguide, respectively, and H′_sst_ = 340 nm. The distance of the adjacent SST films and the increment are also g′_n_ = g′_n−1_ +d′ and g_1′_ = 100 nm. The number of SST films in one of the waveguide sides is N′.

In order to clarify the output characteristics of the polarizer, we define the insertion loss (IL) of the TE or TM mode in the two phases as
IL = −10log_10_ (T) (1)

The extinction ratio (ER) between two modes is then defined as
ER = |IL_TM(cry)_ − IL_TE(cry)_|(2)
where T is the transmittance of the output port of the waveguide for the TE or TM mode.

### 2.2. Light Propagation Simulation

COMSOL Multiphysics was used to simulate the light propagation characteristics of the TM and TE mode of polarizers. The refractive indexes of SST were obtained using spectroscopic ellipsometry as shown in Figure 2. The crystalline phase SST is polycrystalline structure, corresponding to hexagonal phase. At 1550 nm, those refractive indexes of the hybrid Si waveguide and PCMs were chosen from Refs. [27,34] and extracted from Figure 2, listed in Table 1. The scattering parameters were realized by defining the port of the waveguide input as a numerical type. With the default physics-controlled mesh method, the frequency-domain method was applied to calculate the electric field transmission of the waveguide.

### 2.3. Modal Analysis

A boundary-mode analysis provided in COMSOL Multiphysics was applied to calculate the mode profile. The effective refractive index of the hybrid waveguide with the single TE and TM modes was obtained for different states of SST. The effective index of the hybrid waveguide was written as (neff = neff + k_eff_ i).

## 3. Result and Discussion

### 3.1. Optimization of the Structure of the TM-Pass/TE-Pass Polarizers

In order to optimize the structure of the TM-pass polarizer, the ILs with different sizes of H_sst_, L_sst_, N, and d together with both phases of SST were calculated and the results are shown in Figure 3. As for the SST array, the electron beam lithography (EBL) process can be used to form the deposition window for the SST layer. The small gap size of ~5 nm can be achieved by EBL [35]. Considering the flexibility of the fabricate, we set the step of d and L_sst_ to 10 nm. The SST thin films can be sputtered onto the Si waveguide layer by magnetron sputtering. The thin film thickness using magnetron sputtering can be as small as 1 nm [36]. We set the step of H_sst_ to 5 nm. A similar trend of the first-rise-and-then-drop was found for the IL with the increase above four parameters. For the TM-pass polarizer, the unwanted TE mode needs a high IL with c-SST while the other three cases are a low loss. Therefore, we choose the above four parameters as follows: H_sst_ = 50 nm, L_sst_ = 200 nm, d = 10 nm, and N = 5, according to the maximum value in Figure 3a–d. Similarly, the optimized structure parameters of the TE-pass polarizer are with W′_sst_ = 55 nm, L′_sst_ = 225 nm, d′ = 30 nm, and N′ = 5.

### 3.2. Light Propagation Behavior of the TM-Pass/TE-Pass Polarizers

Figure 4 and Figure 5 show the simulated light propagation behavior of the TM- and TE-pass polarizers, demonstrated by normalized electric field transmission profiles. When SST is in the amorphous state, both the TE and TM modes propagation present low losses owing to a low refractive index of SST close to that of the Si waveguide as shown in Figure 4a and Figure 5a. For the TM-pass polarizer, when the SST is in the lossy crystalline phase, the unwanted TE mode is blocked because of a large effective extinction coefficient (k_eff_) resulting in strong attenuation. At this time, the TM mode however can transmit through the hybrid waveguide, thus the polarizer shows a “TM-pass” effect (see Figure 4b).

On the contrary, the TM mode in the TE-pass polarizer will be coupled to the phase-change material on both sides of the hybrid waveguide, thus being cut off in the crystalline state, whereas the TE mode still goes through the waveguide directly, thus the polarizer shows a “TE-pass” effect (see Figure 5b).

### 3.3. Modal Analyses

In order to further explore the working feature of polarizers, we calculated the mode distributions of two polarizers at the hybrid waveguide. Figure 6 shows the electric field mode distribution for TE and TM modes along the propagate direction with different phases of the TM-pass polarizer at the cross-section. At 1550 nm, for the amorphous phase, the refractive index of SST is similar to that of Si (~3.478), so both the TE and TM modes are bound in the Si waveguide in Figure 6a. For the crystalline phase, obvious change of the waveguide mode profiles and the complex effective index of the TE mode can be observed once SST changes from amorphous to the crystalline state, indicating the modulation effect of refractive index and absorptivity of SST. On the contrary, the TM mode is almost unchanged and it is restricted in the waveguide because of a low complex effective refractivity (see Figure 6b). Thus, the SST on the top of the waveguide will affect the TE mode more extensively than the TM mode.

Figure 7 shows the electric field mode distribution of the TE and TM modes along the propagate direction of the TE-pass polarizer at 1550 nm in different phase states. For the TM mode, numerous mode profiles enter into the side SST films once its phase is triggered to the crystalline state, the enhanced SST-light interaction induced the increase in the propagation loss. As for the TE mode, the electric field propagates in the horizontal direction, it is still concentrated in the silicon waveguide when the SST is transformed into a crystalline state, resulting in a weaker interaction between SST and light. Therefore, the effect of SST phase transition on TM mode is more significant than that on TE mode in TM-pass polarizer.

### 3.4. Comparison with GST-Based TM-/TE-Pass Polarizer

To compare the performance of the proposed devices with that of the GST-based ones, we further calculate the ILs and ERs of the GST-based polarizers with a constant geometry.

Figure 8 shows the spectral response of the TM-/TE-pass polarizer based on SST and GST from 1525 nm to 1575 nm, respectively. In Figure 8a,b, for the SST-based TM-pass polarizer, when SST is in the crystalline phase, the IL curve of TE mode exhibits a trend of first increasing and then decreasing with the change of wavelength. The maximum value of 43.1 dB was achieved at 1550 nm, while the IL of TM mode kept below 2.8 dB over the entire wavelength range (1525~1575 nm) (see Figure 8a). For the SST-based TE-pass polarizer, the IL of TM mode in the crystalline was 46.38 dB at 1550 nm, while the IL of TE mode has been kept below 2.2 dB as a whole (see Figure 8b). In the amorphous, the ILs of two polarizers were kept below 2.3 dB in the range of working bandwidth. In Figure 8c,d, as for the GST-based TM- and TE-pass polarizers, when GST is in the crystalline phase, the ILs of the unwanted mode were only 10.6 dB and 14.5 dB at 1550 nm, respectively. While the ILs of the wanted mode kept below 3.3 dB and 2.2 dB overall, respectively.

In general, the amorphous state of SST or GST shows a low value of IL for both the TM-pass and TE-pass polarizers because of an inherent property of phase-change materials with a relatively low extinction coefficient (k). While for the case of crystalline state, used for the “pass” mode in the present design, a relatively low IL was found with c-SST for the above two polarizers, indicating that SST based polarizers have a low loss in the “pass” mode, as compared to those with c-GST. The low value of IL obtained in the SST-based polarizers is possibly caused by the big difference of refractive index between its crystalline and amorphous states. As for the unwanted mode, the ILs of GST-based polarizers are smaller than above SST-based ones. Therefore, it is conducive to realizing a large extinction ratio in these SST-based polarizers.

Figure 9 compares the ER of polarizers based on GST and SST between 1525 nm and 1575 nm. It can be seen that the SST-based polarizers have higher ERs than GST-based in the entire wavelength from 1525–1575 nm. Both the highest points in Figure 9a,b were at 1550 nm. For the SST-based TM-pass polarizer, a high ER of ~43.12 dB was performed at the communication wavelength of 1550 nm, which is 33 dB larger than that of the GST-based polarizer. Similarly, for the SST-based TE pass-polarizer, a high ER of ~44.21 dB was also obtained at 1550 nm, which is 32 dB larger than that of the GST-based one. In Figure 9b, as for the GST-based TE-pass polarizer, it can be seen that the ERs (black line) were maintained at 10.6~15.2 dB from 1525 to 1575 nm. The ER of 15.2 dB for the GST-based TE-pass polarizer was the highest value at 1565 nm, while that of SST-based (31.5 dB) was still larger than the former GST-based counterpart. The SST-based polarizers have the larger ER due to the significant discrepancy in IL between the two modes of TE and TM with the crystalline SST, as shown in Figure 8.

### 3.5. Discussion

Good performances of a polarizer include low insertion loss [14,28,34], high ER [28,37,38], compactness [37], broadband operation [38], and so on. To achieve the characteristic of low loss and high ER, two main methods are described as follows: (1) to change the phase-change material [14] and (2) to design the polarizer structure [28]. In our research, the picosecond-phase-change material was used in the design of polarizers, and we demonstrated two polarizers for either the TM or TE pass with two states of SST, respectively. Our optical polarizers could perform the selective polarization of the TM or TE mode in the crystalline state. In general, the polarizers based on the Si waveguide have considerable size [37,38], whereas those polarizers based on Si-PCMs hybrid waveguide can improve the compactness because adding the PCM can effectively shorten the size of the coupling region [14,28]. In this paper, a small scale only 10 µm-long with several SST bars was designed in the proposed polarizers. Finally, the response time of a polarizer, a significant factor to be considered for those dynamics switching devices, depends on the phase-change time of PCMs between the crystalline and amorphous states. Compared with the traditional phase-change material of GST, SST has an ultrafast response time up to ~700 ps [29], which provides a possibility for the development of ultrafast picosecond photonic chips.

## 4. Conclusions

In summary, two kinds of polarizers based on a phase-change material of SST and Si waveguide were proposed to realize the polarization selectivity of the wanted TM or TE mode. A small scale of 10 µm-long Si waveguide hybrid SST array with a varied distance was designed, which is conducive to on-chip integration. The influence of two SST states of amorphous and crystalline on the mode profiles and light propagation behavior were studied using COMSOL Multiphysics. For the SST-based TM- and TE-pass polarizers, the high ER of ~43.12 dB and ~44.21 dB were performed at the communication wavelength of 1550 nm, respectively. Comparing the proposed two polarizers when the phase-change material was crystalline, SST was more conducive to achieving low-loss performance than GST. (The ERs were 33 dB and 32 dB larger than that of GST-based ones with a constant geometry, respectively). The design of the ultra-high ER, non-volatile TM- and TE-pass polarizers may pave the way for the selective optical polarization devices.

## Figures and Tables

**Figure 1 micromachines-13-00495-f001:**
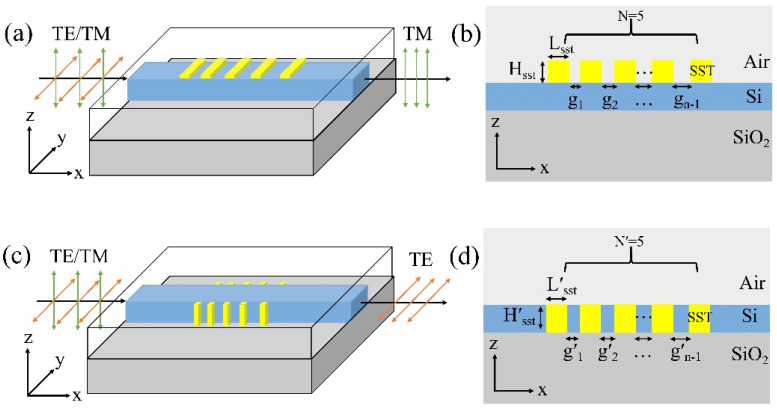
(**a**) The three-dimensional structure schematic of the TM-pass polarizer and its (**b**) cross-section. (**c**) Schematic of the TE-pass polarizer and its (**d**) cross-section.

**Figure 2 micromachines-13-00495-f002:**
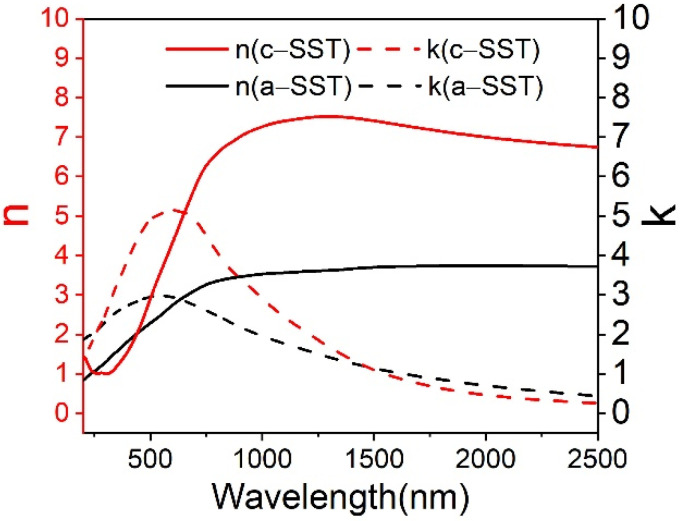
The real (n) and imaginary (k) parts of SST refractive indexes for the amorphous and crystalline state.

**Figure 3 micromachines-13-00495-f003:**
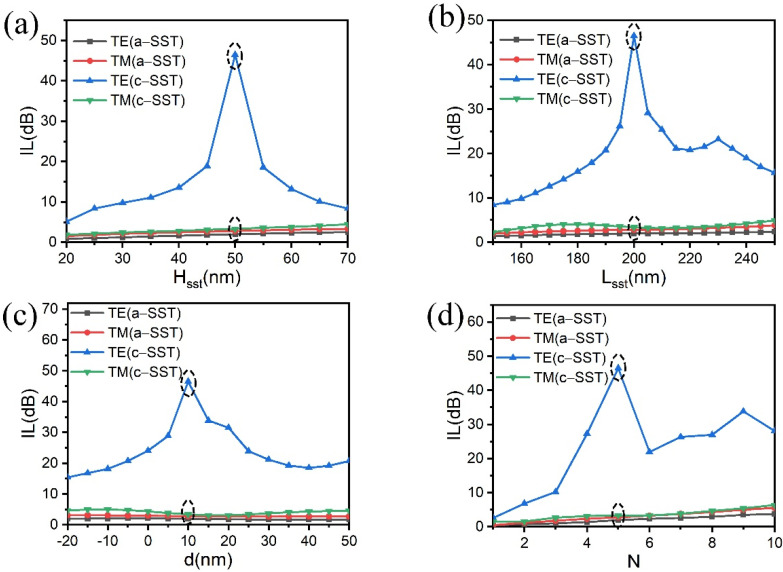
Effect of the (**a**) height of SST H_sst_, (**b**) length of SST L_sst_, (**c**) the increment d of the gap, and (**d**) number N of the SST thin films on the top of the Si waveguide.

**Figure 4 micromachines-13-00495-f004:**
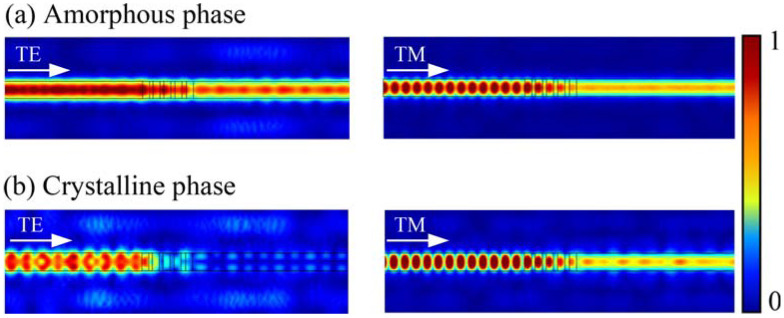
The light propagation simulations of the TM-pass polarizer with different polarization modes. (**a**) When the SST is in the amorphous phase and (**b**) crystalline phase.

**Figure 5 micromachines-13-00495-f005:**
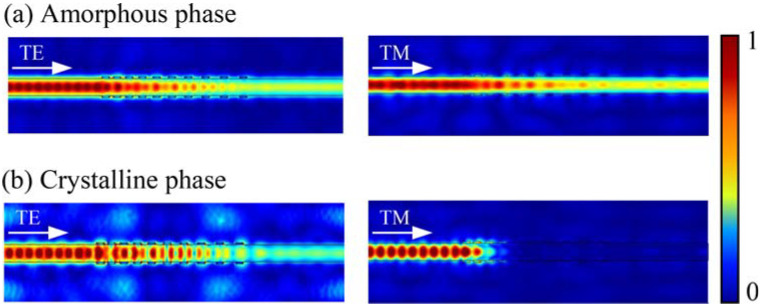
The light propagation simulations of the TE-pass polarizer with different polarization modes. (**a**) When the SST is in the amorphous phase and (**b**) crystalline phase.

**Figure 6 micromachines-13-00495-f006:**
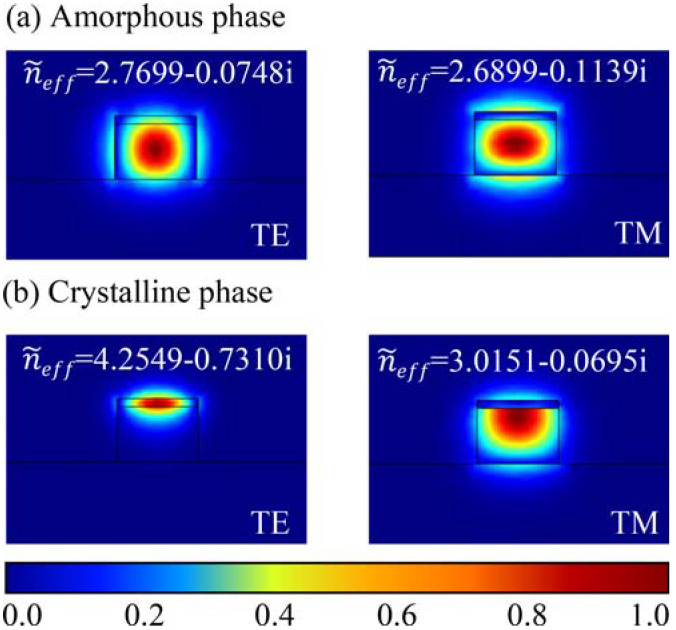
The mode profiles of TE and TM modes of the TM-pass polarizer with SST.

**Figure 7 micromachines-13-00495-f007:**
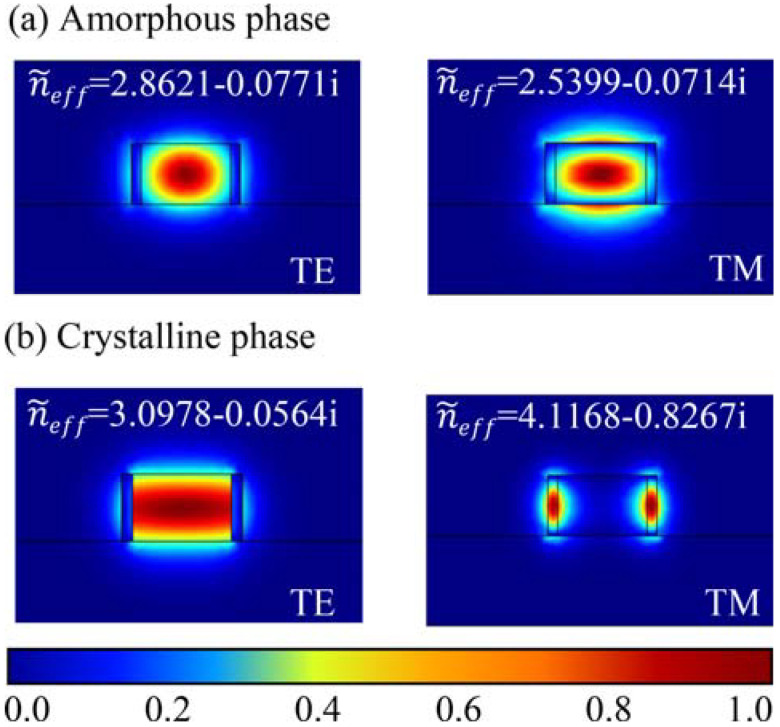
The mode profiles of TE and TM modes of the TE-pass polarizer with SST.

**Figure 8 micromachines-13-00495-f008:**
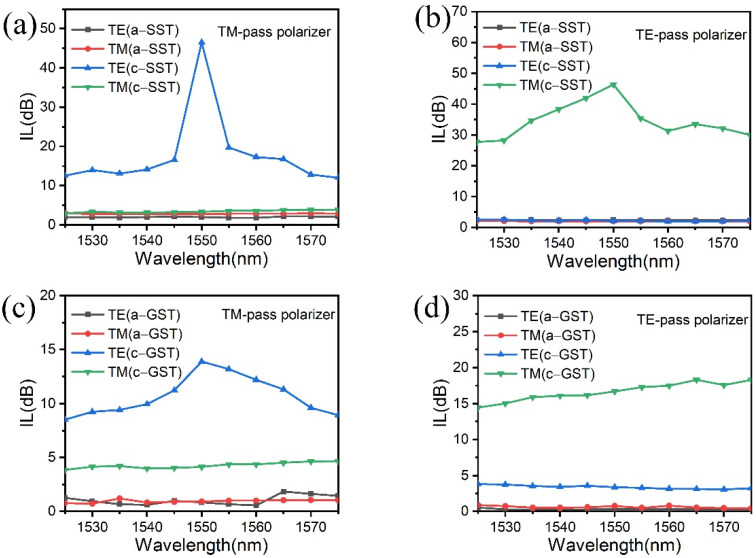
The insertion losses of polarizers based on SST and GST for TE and TM polarization in different states in the range of 1525~1575 nm wavelength. (**a**,**b**) The TM- and TE-pass polarizers based on SST. (**c**,**d**) The TM- and TE-pass polarizers based on GST.

**Figure 9 micromachines-13-00495-f009:**
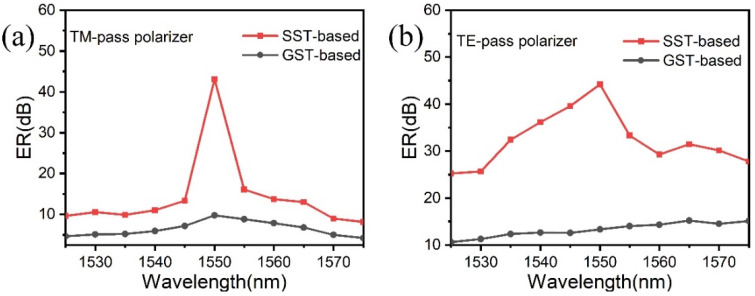
ER comparisons of the TM- and TE-pass polarizers based on different phase-change materials of SST and GST in the range of 1525~1575 nm wavelength. (**a**) The TM-pass polarizer and (**b**) the TE-pass polarizer.

**Table 1 micromachines-13-00495-t001:** Optical material parameters of hybrid waveguide for the simulation.

Material Type	SiO_2_ [27]	Si [34]	a-GST [27]	c-GST [27]	a-SST	c-SST
Refractive index	2.445	3.478	4.6 + 0.12i	7.45 + 1.49i	3.7 + 1.1i	7.3 + 0.9i

The above refractive index is at 1550 nm wavelength.

## Data Availability

Not applicable.

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
