# Peer review of "Design of Ultra-High Extinction Ratio TM- and TE-Pass Polarizers Based on Si-Sc0.2Sb2Te3 Hybrid Waveguide"

_micromachines, 2022, doi:10.3390/mi13040495_

Round 1

Reviewer 1 Report

The manuscript presents a number of simulations for light propagation through SST waveguides for TE and TM modes in the telecom spectral range around 1550 nm. Optical properties of the polarizing waveguides have been optimized using several parameters that influence the insertion loss (IL). They are listed in text and some of them are presented in FIG. 2 (a,b,c,d). It is not clear if the “best values” for each parameter represent the global optimum, or in contrast, each of them has a local optimum, while other parameters are not simultaneously optimized in the multi-dimensional parameter space. Authors should also discuss the ranges for parameter optimization in FIG.2 in terms of technological limits and requirements for miniaturization. For example, can parameter d in FIG.2( c) be as small as 10 nm using current technology?  And, what is the physical meaning of the negative increment of the gap d in FIG.2( c) ?  

Besides, the scientific value of all presented results depend on the input refractive index of SST. So, Table 1 reports the refractive index of c-SST to be 7.3+0.9i and one can assume that this value was obtained from the ellipsometry measurements for continuous SST film as described in Section 2 Experimental. But the same core group of co-authors recently reported measurements of refractive index of c-SST to be very different 5.95+2.95i at the same wavelength of 1550 nm [Xuanxuan Xie et al 2021 J. Phys.: Conf. Ser. 1907 012051]. There is no single comment in the manuscript about this significant discrepancy. It creates a strong suspicion that the new and convenient numbers of refractive index (7.3+0.9i) for c-SST were chosen to obtain better simulation results for insertion loss due to the smaller imaginary part of the refractive index. Thus, the whole story about “excellent” refractive index of SST cannot be trusted. Probably if authors used their previous values of c-SST of 5.95+2.95i  then there won’t be any advantage of using SST compared to GST?  Thus, the manuscript results cannot be trusted and the results cannot be published in the current form.

There are other multiple technical deficiencies listed below.

Abstract

Line 14:  unnecessary use of “novel”. The “novelty” of SST for integrated circuits has been reported previously by the same core authors in [Xuanxuan Xie et al 2021 J. Phys.: Conf. Ser. 1907 012051].

Line 16:  unnecessary use of “excellent”. The non-scientific term “excellent” should be replaced with more specific information about the corresponding refractive index of SST that can be trusted.

Line 17: the abbreviation ER is not explained as extinction ratio

Line 18 “the transmittance behavior in the two polarizers, was further demonstrated in detail.” – such statement is very unclear, especially in the Abstract that should describe specifics.

Line 22:  “The present study provides guidance for the design of high-performance polarizers …” this is not clear. Does this manuscript report the “design” (as is written in the title) or this manuscript is just a “guidance for the design” ?

Manuscript:

Line 57:  "SST" is not defined in the text of the Manuscript (only in the Abstract).

Line 65: “Finally, the properties of the polarizers proposed were also discussed with the previous reports” – this is not a clear sentence.

Starting with the line 75 the manuscript became very hard to read. For example,

Line 76: “The SST arrays with several groups of the film..” not clear what are those “groups of the film”.

Line 80: “The width of SST Wsst has the same width …” should be rewritten.

Line 81:  “the number of the SST film” not clear, the number N is not indicated in Fig. 1a and is not clearly explained in the text.

Author Response

Dear Referees,

Thank you for your careful reading and valuable comments in detail. Those comments are very helpful for revising and improving our paper. We have studied your comments carefully and we hope the revised manuscript can meet with your approval.

We make revision point by point following your decision letter as follow:

(Texts in black are your decisions, and others in red are the responses. The reference in the response is presented by [R])

Point 1: The manuscript presents a number of simulations for light propagation through SST waveguides for TE and TM modes in the telecom spectral range around 1550 nm. Optical properties of the polarizing waveguides have been optimized using several parameters that influence the insertion loss (IL). They are listed in text and some of them are presented in FIG. 2 (a,b,c,d). (a)It is not clear if the “best values” for each parameter represent the global optimum, or in contrast, each of them has a local optimum, while other parameters are not simultaneously optimized in the multi-dimensional parameter space. (b)Authors should also discuss the ranges for parameter optimization in FIG.2 in terms of technological limits and requirements for miniaturization. For example, can parameter d in FIG.2( c) be as small as 10 nm using current technology? (c) And, what is the physical meaning of the negative increment of the gap d in FIG.2( c) ? 

Response 1:

---Thanks for the referee’s comment.

---(a) In Fig.2a-d, the “best values” for each parameter represent the global optimum. In the parameter optimization process, we did global optimization of the parameters, taking TM-pass polarizer for example, we first calculated all combinations of the following parameters: the height of Sc0.2Sb2Te3(SST) (Hsst), the length of SST (Lsst), increment of the gap (d). Considering the actual fabrication flexibility, we choose the first gap (g1) as g1=100 nm [32,33] and the N=5. A set of optimal values were chosen as a trade-off, and then further explored the effect of N, the ranges and calculation steps are shown in the Table R1. The same method was done for the TE-pass polarizer. In the manuscript, the plotted variation pattern of the parameters was obtained from the global optimization. Relevant supplements have been added to the revised manuscript in Line 78-79, 85 (Page 2) in red.

Table R1. Global optimization range of parameters

Parameter

Range (start, step, stop)

Hsst

(20nm,5nm,60nm)

Lsst

(150nm,10nm,350nm)

d*

(-20nm,10nm,20nm)

N

(1,1,10)

*The negative value of d means that gap is gradually decreasing.

---(b) Thank you for your suggestion. we have discussed the parameters of the SST array in the polarizer in Figure 2 according to the fabricate technology. “As for the SST array, the electron beam lithography (EBL) process can be used to form the deposition window for the SST layer. The small gap size of ~5 nm can be achieved by EBL [35]. Considering the flexibility of the fabricate, we set the step of d and Lsst to 10 nm. The SST thin films can be sputtered onto the Si waveguide layer by magnetron sputtering. The thin film thickness using magnetron sputtering can be as small as 1 nm [36]. We set the step of Hsst to 5 nm.” The above description in italic has been added in the revised manuscript in Line 116-121 (Page 3) in red.

---(c)In Figure 2, d is the increment of gap in gn=gn-1+d (when d<0, represent that the gn is decreasing relative to gn-1).

Point 2: Besides, the scientific value of all presented results depend on the input refractive index of SST. So, Table 1 reports the refractive index of c-SST to be 7.3+0.9i and one can assume that this value was obtained from the ellipsometry measurements for continuous SST film as described in Section 2 Experimental. But the same core group of co-authors recently reported measurements of refractive index of c-SST to be very different 5.95+2.95i at the same wavelength of 1550 nm [Xuanxuan Xie et al 2021 J. Phys.: Conf. Ser. 1907 012051]. There is no single comment in the manuscript about this significant discrepancy. It creates a strong suspicion that the new and convenient numbers of refractive index (7.3+0.9i) for c-SST were chosen to obtain better simulation results for insertion loss due to the smaller imaginary part of the refractive index. Thus, the whole story about “excellent” refractive index of SST cannot be trusted. Probably if authors used their previous values of c-SST of 5.95+2.95i  then there won’t be any advantage of using SST compared to GST?  Thus, the manuscript results cannot be trusted and the results cannot be published in the current form.

Response 2:

--- Thanks for your suggestion.

In our previous report, we studied the refractive index of c-SST in FCC phase (FCC-SST) [R2]. In this article, we measured the refractive index of c-SST in HEX phase (HEX-SST). Relevant supplements have been added to the manuscript in Line 101 (Page 3) in red.

SST has been proved to exist at FCC and HEX phases[R1], and our preliminary work focused on the refractive index of FCC-SST and its application in optical switching. However, the cubic phase is the metastable state of hexagonal. In the previous studies, not only ST [R3], but also its compounds doped with C [R4], SiC [R5], Ti [R6], Bi [R7], focused mainly on the stable HEX phase. SST is a compound of Sb2Te3 (ST) doped with Sc, in which the Sb atoms are replaced by Sc atoms in HEX structure of ST [R8] Therefore, here we also focus on the refractive index of the HEX-SST and its application in optical polarization devices. At the same time, n of the HEX-SST is larger than that of the FCC-SST, beneficial to a stronger coupling with SST in hybrid waveguide. k of the HEX-SST is smaller than that of the FCC-SST, which is conductive to reduce the light absorption in waveguide. Therefore, HEX-SST has the excellent characteristics of the refractive index.

There are other multiple technical deficiencies listed below.

Abstract

Point 3: Line 14:  unnecessary use of “novel”. The “novelty” of SST for integrated circuits has been reported previously by the same core authors in [Xuanxuan Xie et al 2021 J. Phys.: Conf. Ser. 1907 012051].

Response 3: Thanks for your suggestion. We would like to express that the optical polarization devices are based on a novel phase change material. To avoid misunderstanding, we have removed the word-”novel”.

Point 4: Line 16:  unnecessary use of “excellent”. The non-scientific term “excellent” should be replaced with more specific information about the corresponding refractive index of SST that can be trusted.

Response 4: Thank you for your concern. We have given a description above, n of the HEX-SST is larger than that of the FCC-SST, beneficial to a stronger coupling with SST in hybrid waveguide. k of the HEX-SST is smaller than that of the FCC-SST, which is conductive to reduce the light absorption in waveguide. Thus, the HEX-SST in this article has a better refractive index than the FCC-SST, so the SST exhibits excellent characteristics.

Point 5: Line 17: the abbreviation ER is not explained as extinction ratio

Response 5: According to review’s comment, we have added “extinction ratio” to explain the abbreviation ER in Line 17 (Abstract) in red

Point 6: Line 18 “the transmittance behavior in the two polarizers, was further demonstrated in detail.” – such statement is very unclear, especially in the Abstract that should describe specifics.

Response 6: Thanks for your suggestion. We have modified this sentence and describe the transmission behavior of the polarizer in detail as well as the transmission behavior of TE and TM modes in the two polarizers, was further demonstrated in detail. when the SST is crystalline, the unwanted mode can be attenuated, while the wanted mode can pass through with low loss.in Line 18-21(Abstract) in red.

Point 7: Line 22:  “The present study provides guidance for the design of high-performance polarizers …” this is not clear. Does this manuscript report the “design” (as is written in the title) or this manuscript is just a “guidance for the design” ?

Response 7: Thank you for your comment. This manuscript reports the design of the polarizer. For more clarity and correct description, we have modified the sentence to “The design of high-performance polarizers paves a new way for applications of all-optical integrated circuits.” in Line 23-24 (Abstract) in red.

Manuscript:

Point 8: Line 57:  "SST" is not defined in the text of the Manuscript (only in the Abstract).

Response 8: According to review’s comments, we have added “Sc0.2Sb2Te3” to define the SST in Line 58 (Page 2) in red.

Point 9: Line 65: “Finally, the properties of the polarizers proposed were also discussed with the previous reports” – this is not a clear sentence.

Response 9: Thanks for your comment. We have changed the sentence to a more explicit description—“Finally, the properties of the polarizers proposed were also compared with the GST-base ones”in Line 66-67 (Page 2) in red.

Starting with the line 75 the manuscript became very hard to read. For example,

Point 10: Line 76: “The SST arrays with several groups of the film.” not clear what are those “groups of the film”.

Response 10: Thanks for your suggestion. To make it easier to understand, we changed the description to “The five bars of SST thin films…” in Line 71 (Page 2) in red.

Point 11: Line 80: “The width of SST Wsst has the same width …” should be rewritten.

Response 11: Thanks for your concern. We modified the sentence to “The width of SST thin film is 500 nm.…” in Line 75 (Page 2) in red.

Point 12: Line 81:  “the number of the SST film” not clear, the number N is not indicated in Fig. 1a and is not clearly explained in the text.

Response 12: Thanks for your suggestion. We have indicated the N=5 and N’=5 in Fig.1, and added “The five bars of SST thin films…” in the text in Line 71 and Line 80 (Page 2) in red.

References

  1. Xu, Z.; Lyu, T.; Sun, X. Interleaved Subwavelength Gratings Strip Waveguide Based TM Pass Polarizer on SOI Platform. IEEE Photonics J. 2020, 12, doi:10.1109/jphot.2020.2968570.
  2. Yu, W.; Dai, S.; Zhao, Q.; Li, J.; Liu, J. Wideband and compact TM-pass polarizer based on hybrid plasmonic grating in LNOI. Optics Express 2019, 27, 34857-34863, doi:10.1364/oe.27.034857.
  3. Duan, H.; Hu, H.; Kumar, K.; Shen, Z.; Yang, J.K.W. Direct and Reliable Patterning of Plasmonic Nanostructures with Sub-10-nm Gaps. Acs Nano 2011, 5, 7593-7600, doi:10.1021/nn2025868.
  4. Huang, Y.; Liu, F.; Zhang, Y.; Li, W.; Han, G.; Sun, N.; Liu, F. Effects of biaxial strain on interfacial intermixing and local structures in strain engineered GeTe-Sb2Te3 superlattices. Appl. Surf. Sci. 2019, 493, 904-912, doi:10.1016/j.apsusc.2019.07.069.

 R1. Rao, F.; Ding, K.Y.; Zhou, Y.X.; Zheng, Y.H.; Xia, M.J.; Lv, S.L.; Song, Z.T.; Feng, S.L.; Ronneberger, I.; Mazzarello, R.; et al. Reducing the stochasticity of crystal nucleation to enable subnanosecond memory writing. Science 2017, 358, 1423-1426, doi:10.1126/science.aao3212.

R2. Xie, X.; Liu, F.; Zhang, L.; Lian, Y.; Li, Y. Ultrafast non-volatile 1x1 optical switch using phase change material Sc2Sb2Te3. Journal of Physics: Conference Series 2021, 1907, 012051, doi:10.1088/1742-6596/1907/1/012051.

R3. Liu, T.; Deng, H.; Cao, H.; Zhou, W.; Zhang, J.; Liu, J.; Yang, P.; Chu, J. Structural, optical and electrical properties of Sb2Te3 films prepared by pulsed laser deposition. Journal of Crystal Growth 2015, 416, 78-81, doi:10.1016/j.jcrysgro.2015.01.022.

R4. Meng, Y.; Behera, J.K.; Wen, S.; Simpson, R.E.; Shi, J.; Wu, L.; Song, Z.; Wei, J.; Wang, Y. Ultrafast Multilevel Optical Tuning with CSb2Te3 Thin Films. Advanced Optical Materials 2018, 6, doi:10.1002/adom.201800360.

R5. Meng, Y.; Wu, L.; Song, Z.; Wen, S.; Jiang, M.; Wei, J.; Wang, Y. Silicon carbide doped Sb2Te3 nanomaterial for fast-speed phase change memory. Materials Letters 2017, 201, 109-113, doi:10.1016/j.matlet.2017.05.003.

R6. Zhu, M.; Xia, M.; Rao, F.; Li, X.; Wu, L.; Ji, X.; Lv, S.; Song, Z.; Feng, S.; Sun, H.; et al. One order of magnitude faster phase change at reduced power in Ti-Sb-Te. Nature Communications 2014, 5, doi:10.1038/ncomms5086.

R7. Tongpeng, S.; Sarakonsri, T.; Isoda, S.; Haruta, M.; Kurata, H.; Thanachayanont, C. Electron Microscopy investigation of Sb2-xBixTe3 hexagonal crystal structure growth prepared from sol-gel method. Materials Chemistry and Physics 2015, 167, 246-252, doi:10.1016/j.matchemphys.2015.10.039.

R8. Hu, S.; Liu, B.; Li, Z.; Zhou, J.; Sun, Z. Identifying optimal dopants for Sb2Te3 phase-change material by high-throughput ab initio calculations with experiments. Computational Materials Science 2019, 165, 51-58, doi:10.1016/j.commatsci.2019.04.028.

Reviewer 2 Report

This manuscript develop polarizers based on SST silicon waveguide. The results show high extinction ratios, compare with the polarizers on GST waveguide. However, the authors should address the following comments.

  1. The experiments show nice results.  There is any novel in the design of the polarizers.The authors should explain  the reason in detail.
  2. The authors should put Fig.7 and Fig.8 together for the comparison.
  3. In Fig.9, the highest point is near the 1550 nm for the polarizers of TM  and of TE on SST. However, it is near the 1565 nm of TE on GST? 

Author Response

Dear referees,

Thank you for your careful reading and valuable comments in detail. Those comments are very helpful for revising and improving our paper. We have studied your comments carefully and we hope the revised manuscript can meet with your approval.

We make revision point by point following your decision letter as follow:

(Texts in black are your decisions, and others in red are the responses.)

Reviewer: This manuscript develop polarizers based on SST silicon waveguide. The results show high extinction ratios, compare with the polarizers on GST waveguide. However, the authors should address the following comments.

Point 1:The experiments show nice results.  There is any novel in the design of the polarizers. The authors should explain  the reason in detail.

Response 1:Thanks for your comment. The innovations of this work are as follows:

  1. The polarizer device is based on a novel non-volatile ultrafast phase change material Sc0.2Sb2Te3(SST), and SST has a better refractive index than GST.
  2. The structure of the polarizers are based on the Si waveguide and the unequal spacing phase change material array. In previous studies, the polarizers were based on equally spaced arrays.
  3. The SST based polarizer has a higher extinction ratio than the GST based polarizer.

Point 2:The authors should put Fig.7 and Fig.8 together for the comparison.

Response 2:Thanks for your suggestion. We have taken your suggestion to put the Fig.7 and Fig.8 together. We have also modified the comparative analysis accordingly. We added the description of the GST-based polarizer at 1550 nm in Line 189-191(Page 6) in red.

Point 3:In Fig.9, the highest point is near the 1550 nm for the polarizers of TM  and of TE on SST. However, it is near the 1565 nm of TE on GST?

Response 3:Thanks for your concern. In Fig.9 (Now in Fig.8), the extinction ratio of TE-pass polarizer based on GST has a highest point near the 1565nm, this is due to the 1565 has the largest insertion loss for the unwanted TM mode. However, the ER of the GST-based polarizer is much smaller than the ER of the SST-based at 1565 nm. In addition, our study focuses on the communication wavelength of 1550 nm, which is a common wavelength for optical devices.

Round 2

Reviewer 1 Report

Most of my previous recommendations has been included, but the most

important question about the refractive index of SST remains open. Now authors confused me even more writing that 

their c-SST is in the “HEX phase”, probably meaning that SST is in the

hexagonal phase. But then we immediately have a problem, because every

hexagonal material has two (!) values of the refractive index. One for E-field

along the hex axis and a different one for the perpendicular direction. For most of the grown 

hexagonal materials the direction of the axis is perpendicular 

to the substrate. Thus, the same refractive index values cannot be used in simulations

for both TE and TM modes as it appears in the Table. Or, maybe,

the authors think that the SST  is in a polycrystalline phase and the refractive index

number they used is the average for two orthogonal

directions? In any case, this information should be clarified before the manuscript can

be published. Otherwise, the whole set of the presented simulations is based on

unjustified value of the refractive index. BTW, if the authors really have a new set of 

ellipsometry data for SST, then such data must be shown in the text, or Appendix,

or Supplement. My version of the revised

manuscript does not have such information.

Author Response

Dear Referees,

Thank you for your careful reading and valuable comments in detail. Those comments are very helpful for revising and improving our paper. We have studied your comments carefully and we hope the revised manuscript can meet with your approval.

We make revision point by point following your decision letter as follow:

(Texts in black are your decisions, and others in red are the responses.)

Point 1:Most of my previous recommendations has been included, but the most important question about the refractive index of SST remains open. Now authors confused me even more writing that their c-SST is in the “HEX phase”, probably meaning that SST is in the hexagonal phase. But then we immediately have a problem, because every hexagonal material has two (!) values of the refractive index. One for E-field along the hex axis and a different one for the perpendicular direction. For most of the grown hexagonal materials the direction of the axis is perpendicular to the substrate. Thus, the same refractive index values cannot be used in simulations for both TE and TM modes as it appears in the Table. Or, maybe, the authors think that the SST  is in a polycrystalline phase and the refractive index number they used is the average for two orthogonal directions? In any case, this information should be clarified before the manuscript can be published. Otherwise, the whole set of the presented simulations is based on unjustified value of the refractive index. BTW, if the authors really have a new set of ellipsometry data for SST, then such data must be shown in the text, or Appendix, or Supplement. My version of the revised manuscript does not have such information.

Response 1: Thanks for your suggestions. I'm very sorry that our description was not clear enough. As you mentioned, the Sc0.2Sb2Te3 (SST) sample prepared in our experiment is actually polycrystalline phase and the detailed explanation are given below.

Firstly, the SST thin films were prepared by magnetron sputtering, and crystalline SST thin film was obtained by isothermal annealing at 300℃ for 10 min. Fig. R1(a) shows the TEM image of SST film after 300℃ annealing, the obvious grains in tens of nanometers of size can be observed. The inset shows the selected area electron diffraction (SAED) pattern of SST crystalline state, and the obvious diffraction rings correspond to the (0 1 5), (1 1 0) and (1 2 14) planes, respectively. The X-ray diffraction pattern (XRD) curve of the annealed sample is shown in Fig. R1(b). In Fig. R1(b), the characteristic peaks (0 0 6), (1 0 4), (0 1 5), (1 0 10), (0 1 11) (0 0 15), (1 0 1 3) and (1 0 19) matched quite well with Sb2Te3 (PDF# 15-0874). Therefore, from Fig. R1 we can confirm that the SST film was polycrystalline structure with hexagonal (HEX) phase. Moreover, the refractive indexes of SST were also obtained from these samples. Therefore, the measured refractive index of HEX-SST in our manuscript can be used for both TE and TM modes.

According to your suggestion, the refractive indexes of SST were added in the revised manuscript in Page 3 (section 2.2 Light propagation simulation) in red.

Figure R1. (a) The TEM image of the SST film after 300℃ annealing. (b) The X-ray diffraction pattern for the HEX crystalline SST.

Reviewer 2 Report

The authors have addressed my comments. However,  why there is unwanted TM mode only on the 1565 nm?  does it disappear for next measuremt?  The authors should discuss in the manucript. Moreover, 1565nm belong to C band of communication.

Author Response

Dear Referees,

Thank you for your careful reading and valuable comments in detail. Those comments are very helpful for revising and improving our paper. We have studied your comments carefully and we hope the revised manuscript can meet with your approval.

We make revision point by point following your decision letter as follow:

(Texts in black are your decisions, and others in red are the responses)

Point 1: The authors have addressed my comments. However, why there is unwanted TM mode only on the 1565 nm?  does it disappear for next measuremt? The authors should discuss in the manucript. Moreover, 1565nm belong to C band of communication.

Response 1: Thank you for your suggestions. In order to make it clear that figure 9 refers to TM- / TE-pass polarizers, the label in the figures were changed.

In Fig.9, it shows that the ERs of the two polarizers based on SST and GST with wavelength range from 1525 nm to 1575 nm. The ER is the difference between two ILs of TE and TM modes for the crystalline state, as in Eq.2 in manuscript. It can be seen that the SST-based polarizers have higher ERs than GST-based in the entire range from 1525-1575 nm. And the highest point was at 1550nm.

In Fig.9b, no matter for the GST-based or SST-based TE-pass polarizer, the unwanted TM mode is attenuated when PCM is in the crystalline state. As for the GST-based TE-pass polarizer, it can be seen that the ERs (black line) were maintained about 10.6 ~ 15.2 dB from 1525 to 1575 nm. (It is proved that the device realizes the function that TM mode cannot pass, but the TE mode can pass from 1525 to 1575 nm.) However, the highest value ER of 16.3 dB for the GST-based TE-pass polarizer was at 1565 nm. At the same time, the ER of SST-based is 31.5 dB (the second highest point), 15 dB lager than GST-based.

According to your suggestion, we added the discussion in the revised manuscript (Line 213-215, 218-222) in red 

Round 3

Reviewer 1 Report

Authors answered all my questions. I recommend this manuscript for publication.